# The Applicability of the Drought Index and Analysis of Spatiotemporal Evolution Mechanisms of Drought in the Poyang Lake Basin

**Zihan Gui** [1], **Heshuai Qi** [1], **Faliang Gui** [2,*], **Baoxian Zheng** [1], **Shiwu Wang** [1] and **Hua Bai** [2]

1 Zhejiang Institute of Hydraulics and Estuary (Zhejiang Institute of Marine Planning and Design), Hangzhou 310020, China; guizh3@mail3.sysu.edu.cn (Z.G.); qihs0323@126.com (H.Q.); zhbx_zj@163.com (B.Z.); shiwuwang01@outlook.com (S.W.)
2 Nanchang Institute of Technology, Nanchang 330099, China; baihua1985@126.com
* Correspondence: guifaliang@126.com

**Abstract:** Poyang Lake, the largest freshwater lake in China, is an important regional water resource and a landmark ecosystem. In recent years, it has experienced a period of prolonged drought. Using appropriate drought indices to describe the drought characteristics of the Poyang Lake Basin (PLB) is of great practical significance in the face of severe drought situations. This article explores the applicability of four drought indices (including the precipitation anomaly index (PJP), standardized precipitation index (SPI), China Z-index (CPZI), and standardized precipitation evapotranspiration index (SPEI)) based on historical facts. A systematic study was conducted on the spatiotemporal evolution patterns of meteorological drought in the PLB based on the optimal drought index. The results show that SPI is more suitable for the description of drought characteristics in the PLB. Meteorological droughts occur frequently in the summer and autumn in the PLB, with the frequency of mild drought being 17.29% and 16.88%, respectively. The impact range of severe drought or worse reached 22.19% and 28.33% of the entire basin, respectively. The probability of drought occurrence in the PLB shows an increasing trend in spring, while in most areas, it shows a decreasing trend in other seasons, with only a slight increase in the upper reaches of the Ganjiang River (UGR). One of the important factors influencing drought in the PLB is atmospheric circulation. The abnormal variation of the Western Pacific Subtropical High was one of the key factors contributing to the severe drought in the PLB in 2022. This study is based on a long-term series of meteorological data and selects the drought index for the PLB. It describes the spatiotemporal distribution characteristics and evolution patterns of drought and investigates the developmental path and influencing factors of drought in typical years. This study provides a reliable scientific basis for similar watershed water resource management.

**Keywords:** drought characteristics; Poyang Lake; drought index; applicability; developmental path

## 1. Introduction

Drought is a climatic phenomenon characterized by a long-term lack of rainfall or no rainfall, leading to dry air and water-deficient soil [1]. In 1986, the World Meteorological Organization (WMO) defined drought as "a period of abnormally dry weather characterized by sustained and prolonged rainfall deficiency" [2]. Different from the dangers posed by floods, earthquakes, and other disasters, it is a slowly developing hazard. It begins with a shortage of soil moisture, followed by a decrease in river flow, leading to insufficient reservoir storage and the depletion of groundwater levels, ultimately resulting in negative impacts on the economy and society [3,4]. Although the effects of global warming on droughts are not immediate, it is expected that when droughts occur, they may happen more quickly and intensely [5]. In this context, the frequency, extent, and duration of droughts are all showing an increasing trend [6–8]. This poses a significant threat

to global agriculture, water resources, ecological and environmental security, and social sustainable development.

Droughts have complex impacts and are difficult to understand and predict [9]. The development and implementation of drought indexes have addressed this issue by providing a convenient means for qualitative and quantitative measurement of drought events [10]. Drought indexes based on climate data have been widely used worldwide due to their advantages of convenient information acquisition, flexible time scale, and sensitive response to drought monitoring [11–13]. Currently, many different drought indexes are used as drought monitoring tools. The most commonly used indexes include the Palmer Drought Severity Index (PDSI), which is widely used in the United States [14]; the Deciles Index (DI), which is applied in Australia [15]; the China-Z Index (CPZI), which is used by the National Institute of Metrology of China [16]; and the Standardized Precipitation Index (SPI) and Standardized Precipitation Evapotranspiration Index (SPEI), which are widely applied worldwide [17,18]. These indicators can be roughly divided into two categories: one category considers multiple hydrological factors with relatively clear physical mechanisms but high data requirements, such as drought indices like PDSI and SPEI; the other category reflects drought characteristics by studying the statistical distribution patterns of single hydrological factors such as precipitation through statistical methods, such as drought indices like SPI and CPZI. Drought is influenced by various factors such as local climate, soil conditions, vegetation, and more. Different drought indices exhibit inconsistent patterns in analyzing drought dynamics across different spatial and temporal scales [19]. Stefanidis found that based on the SPEI, the frequency of drought events is increasing at shorter time scales, while it is decreasing at longer time scales but with longer durations [20]. Hayes found that the SPI identifies water deficits more quickly than the PDSI and is a more reliable indicator for monitoring drought development in the southern plains and southwestern United States [21]. Tirivarombo found that compared to SPI, SPEI can identify more drought events with longer durations and higher severity levels [22]. Zeybekoğlu found that at long temporal scales, SPI and CPZI exhibit consistent performance in detecting drought events, but they show differences in short-term drought events [23].

China is situated in the East Asian monsoon region. Due to its complex geographical conditions and climate change, China is one of the most vulnerable regions in the world in terms of climate. Meteorological disasters occur frequently within its borders [24,25]. The Poyang Lake Basin (PLB) is one of the key areas globally for water security and biodiversity conservation [26]. It is an important production base for crops, oilseeds, cotton, and aquatic products in China [27]. However, due to the uneven seasonal distribution of precipitation, the PLB has become one of the regions in China most prone to both floods and droughts [28]. Statistics from the past 50 years indicate that drought disasters frequently occur in the basin, causing significant damage to the environment and agricultural development [29–31]. Therefore, in order to effectively monitor and predict the occurrence of drought and minimize socio-economic losses, extensive research has been conducted on drought characteristics. Min [32] studied the climatic evolution characteristics of drought in the PLB over the past 1000 years and found that the region is currently experiencing a period of frequent droughts. Hong [33] investigated the spatiotemporal evolution of drought in the PLB based on the SPI, and the results indicate that drought in the basin had obvious seasonal characteristics, with a noticeable drying trend in spring and autumn. Liu [28] found that the PLB showed a trend of spring drought based on the SPEI, while it generally exhibited a trend of wetness in other seasons. Wang [34] analyzed the reasons for frequent extreme drought events in the PLB surrounding areas over the past 60 years based on the SPEI. Zheng [35] discovered an increase in the frequency of summer and autumn droughts in the PLB based on comprehensive drought indices.

Although many studies have studied the drought characteristics of PLB, these studies have utilized different drought indices and identified varying drought patterns. The inconsistent results obtained from different drought indices suggest that there are issues with the applicability of drought indexes to the watershed. In addition, these studies

were based on a relatively small number of observation stations (e.g., Liu [28] utilized 41 observation stations, Hong [33] utilized 13 observation stations, Wang [34] utilized 23 observation stations). The limited number of stations used cannot provide sufficient data to explore the spatiotemporal evolution of drought, which can lead to uncertain and problematic analysis results. Furthermore, more work is needed to better understand the mechanisms of drought in PLB, particularly the influencing factors of drought and the strategies to mitigate its impact.

The reasonable evaluation of drought indices is the fundamental and most important issue for drought monitoring, forecasting, research, and related activities. In this study, we utilized monthly data from 185 rainfall stations and 85 meteorological stations within PLB from 1960 to 2022. Based on historical facts, we objectively selected appropriate drought indices to characterize the drought characteristics. Building upon the revelation of interannual variations and fundamental spatial patterns of drought at the watershed scale, special emphasis is placed on the spatiotemporal characteristics of seasonal meteorological drought occurrences. Based on the drought characteristics, the development process of meteorological drought at the watershed scale is delineated, and the influencing factors of drought and their interrelationships are explored. The aim of this study is to provide a scientific basis for understanding the regional characteristics of drought occurrence and the evolutionary patterns of drought processes in the PLB under changing environmental conditions. It is essential to conduct a reasonable assessment of the impacts of drought and develop preventive strategies.

## 2. Study Area

Poyang Lake is currently the largest freshwater lake in China, located on the right bank of the middle and lower reaches of the Yangtze River. It is a lake characterized by seasonal fluctuations in water levels [36]. Poyang Lake Basin (PLB) covers an area of 162,225 km$^2$, accounting for 9% of the controlled basin area of the Yangtze River. The maximum length of the lake (north–south direction) is 173 km, while the maximum width (east–west direction) is approximately 74 km. The total length of the lake shoreline is about 1200 km. PLB is surrounded by mountains on the east, south, and west sides, with higher terrain, while the central and northern parts are lower, sloping inward from the periphery. This configuration forms a fan-shaped topography with the Poyang Lake plain as the base opening northward [37]. PLB is crisscrossed by numerous rivers and dotted with lakes and reservoirs. Among them, covering an area of over 10,000 km$^2$, are the five major river systems of the Gan River, Fu River, Xin River, Rao River, and Xiu River (as shown in Figure 1).

The Poyang Lake Basin overlaps with Jiangxi Province of China by 94% in terms of area. Jiangxi Province is prone to frequent drought disasters. According to historical records [38], from the year 807 to 1949, Jiangxi Province experienced a total of 172 widespread drought events, accounting for 15% of the total years, which means an average occurrence of once every 7 years. After 1949, there was an average of one severe drought every 14 years, with localized water and drought disasters occurring almost every year. It is often said that there would be minor disasters every three years and major disasters every five years. From 1990 to 2022, there have been more than 20 widespread drought disasters in Jiangxi Province. Among them, typical drought events occurred in 1991, 2003, 2007, 2009, 2013, 2019, and 2022.

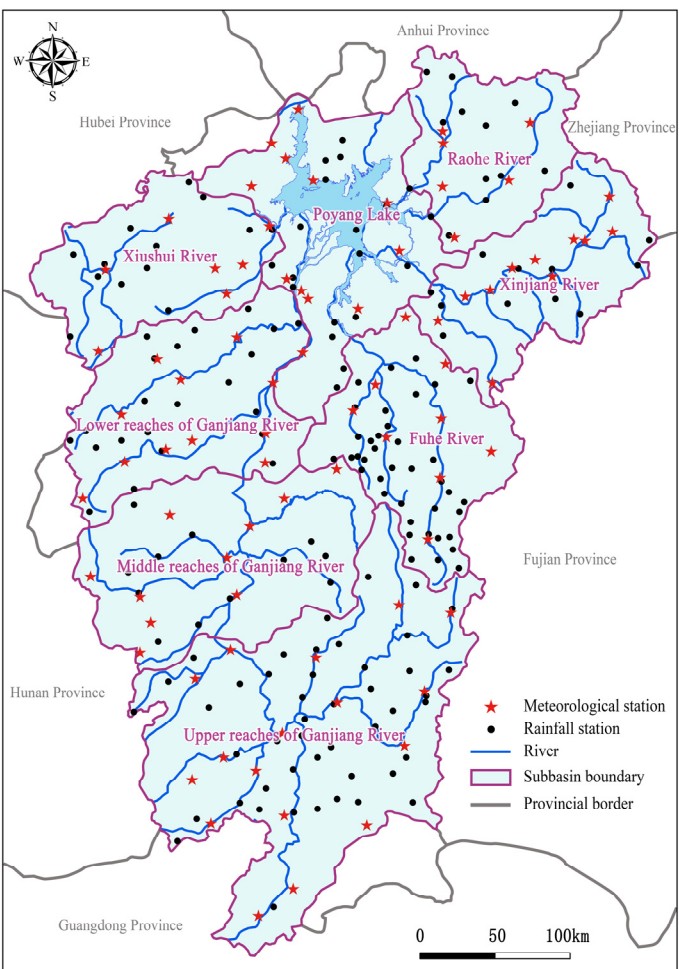

**Figure 1.** Water resource zoning map of the Poyang Lake Basin.

## 3. Selection of Drought Indexes

### 3.1. Commonly Used Drought Indexes

Based on considerations such as the physical mechanisms and computational complexity of various drought indices, this study selects the anomaly percentile index (PJP), SPI, SPEI, and CPZI to systematically analyze the applicability of meteorological and hydrological drought indices in the Poyang Lake Basin.

(1) PJP can be obtained by comparing the precipitation at different time scales with the long-term average of hydrological elements at the same time scale. The calculation formula is as follows:

$$PJP = \frac{P - \overline{P}}{\overline{P}} \times 100\% \tag{1}$$

where P represents the precipitation amount during a certain period, mm, and $\overline{P}$ represents the climatological average precipitation of the corresponding period, mm.

(2) SPI is a meteorological drought index recommended by the World Meteorological Organization for assessing regional meteorological drought conditions. It has evolved into various drought indices characterizing hydrological droughts and agricultural droughts. The Standardized Precipitation Index uses the standardized cumulative frequency distribution of precipitation based on the gamma function to describe precipitation changes [39]. SPI can be calculated using different time scales, such as 1, 3, 6, and 12 months, making it an effective method for reflecting drought conditions in various regions and time periods. The calculation formula is explained below.

The probability density function of the Gamma distribution is as follows:

$$g(x) = \frac{1}{\beta^{\alpha}\gamma(x)}x^{\alpha-1}e^{\frac{-x}{\beta}} \tag{2}$$

where $\alpha > 0$ is the shape parameter, $\beta > 0$ is the scale parameter, and $x > 0$ represents precipitation amount.

The cumulative probability for a given time duration can be calculated using the following equation:

$$G(x) = \int_0^x g(x)dx \tag{3}$$

When $0 < G(x) \leq 0.5$,

$$SPI = -(t - \frac{c_0 + c_1 t + c_2 t^2}{1 + d_1 t + d_2 t^2 + d_3 t^3}) \tag{4}$$

$$t = \sqrt{\ln\left[\frac{1}{(G(x))^2}\right]} \tag{5}$$

When $0.5 < G(x) \leq 1$,

$$SPI = t - \frac{c_0 + c_1 t + c_2 t^2}{1 + d_1 t + d_2 t^2 + d_3 t^3} \tag{6}$$

$$t = \sqrt{\ln\left[\frac{1}{(1 - G(x))^2}\right]} \tag{7}$$

where $c_0 = 2.515517$, $c_1 = 0.802853$, $c_2 = 0.010328$, $d_1 = 1.432788$, $d_2 = 0.189269$, and $d_3 = 0.001308$.

(3) CPZI is an SPI alternative index developed by the China National Climate Center in 1995, which also has multi-scale characteristics. Unlike SPI, CPZI assumes that precipitation follows the P-III distribution, and the calculation formula is as follows:

$$CPZI_{ij} = \frac{6}{C_{si}}\left(\frac{c_{si}}{2}\varphi_{ij} + 1\right)^{1/3} - \frac{6}{C_{si}} + \frac{C_{si}}{6} \tag{8}$$

$$C_{si} = \frac{\sum_{j=1}^{n}\left(x_{ij} - \overline{x_1}\right)^3}{n\sigma_i^3} \tag{9}$$

$$\varphi_{ij} = \frac{x_{ij} - \overline{x_1}}{\sigma_i} \tag{10}$$

$$\sigma_i = \sqrt{\frac{1}{n}\sum_{j=1}^{n}\left(x_{ij} - \overline{x_1}\right)^2} \tag{11}$$

$$\overline{x_1} = \frac{1}{n}\sum_{j=1}^{n}x_{ij} \tag{12}$$

where i represents the given time scale, such as 1, 2, 3, ..., 36 months; j is the current month; $C_{si}$ is the skewness coefficient; n is the total number of months in the study; $\varphi_{ij}$ is the standardized variable; and $x_{ij}$ is the precipitation value for the j month under the i time scale.

(4) Drought is not only influenced by precipitation but is also closely related to evapotranspiration. SPEI is an indicator used to characterize the probability of the difference between precipitation and evapotranspiration over a certain period. It is mainly suitable for monitoring and assessing droughts at different time scales in semi-arid and semi-humid regions. The calculation steps of the SPEI are as follows.

a. Calculating potential evapotranspiration (PET). In this study, we adopt the Thornthwaite method recommended by Vicente-Serrano to calculate potential evapotranspiration. The advantage of this method is that it takes into account temperature variations and provides a better reflection of surface potential evapotranspiration.

b. To calculate the difference $D_i$ between monthly precipitation $P_i$ and potential evapotranspiration $PET_i$,

$$D_i = P_i - PET_i \tag{13}$$

c. Similar to the SPI method, the $D_i$ is normalized, and the corresponding SPEI is calculated. As the original data $D_i$ may contain negative values, a three-parameter log-logistic probability distribution is used for the SPEI.

The drought intensity thresholds for various drought index standards are shown in Table 1.

**Table 1.** Classification of drought intensity levels [17,18,40].

| Level | Type | PJP (Monthly Scale) | SPI/SPEI | CPZI |
|---|---|---|---|---|
| 1 | No drought | $(-40, 0)$ | $(-0.5, 0)$ | $(-0.524, 0)$ |
| 2 | Mild drought | $(-60, -40]$ | $(-1.0, -0.5]$ | $(-1.037, -0.524]$ |
| 3 | Moderate drought | $(-80, -60]$ | $(-1.5, -1.0]$ | $(-1.645, -1.037]$ |
| 4 | Severe drought | $(-95, -80]$ | $(-2.0, -1.5]$ | $(-1.96, -1.645]$ |
| 5 | Extreme drought | $\leq -95$ | $\leq -2.0$ | $\leq -1.96$ |

### 3.2. Analysis of Applicability of Drought Index

Based on monthly precipitation and average monthly temperature data from 190 rainfall stations and 85 meteorological stations within eight water resource tertiary regions in the Poyang Lake Basin, including the Upper Gan River (UGR), Middle Gan River (MGR), Lower Gan River (LGR), Fu River (FR), Xin River (XJR), Rao River (RR), Xiu River (XR), and Poyang Lake area (PYL), the following indices were calculated for each water resource region: PJP, SPI, CPZI, and SPEI. The data covered the period from 1960 to 2020 (the locations of the stations are shown in Figure 1). Based on historical records of typical years of disasters, the differences in representation among the four drought indices in actual drought conditions are compared. The drought index that best represents the actual drought conditions is considered the most optimal for this study based on the intensity and probability of drought occurrence. In this study, 2003 and 2019 were selected as typical years for the analysis of drought index applicability.

Figure 2 shows the representation of the selected four drought indices in describing the drought conditions during the typical year of 2003. Based on the historical records of drought disasters in the Poyang Lake Basin in 2003, the rainfall in July ranked second lowest on record, accounting for only 21.85% of the long-term average for the same period. From late August to early November, the average rainfall in the entire basin was nearly 60% below the long-term average for the same period. Severe droughts were widespread in the southern and western parts of Jiangxi Province, as well as in the PYL, and extended to the middle reaches of the XJR. The persistent high temperatures and lack of rainfall caused the water levels in rivers throughout the province to continue to decline, resulting in a sharp reduction in the storage capacity of various water conservancy projects. Some parts of the UGR, MGR, and FR even broke historical records for the lowest water levels.

Taking July 2003 as an example, in the UGR, the SPI, CPZI, PJP, and SPEI were −3.17, −2.43, −89.03%, and −2.18, respectively. The corresponding evaluations of drought intensity levels were "extreme drought", "extreme drought", "severe drought", and "extreme drought", indicating that the PJP index underestimated the regional drought intensity in this particular case. Similar phenomena also occurred in the MGR (SPI, CPZI, PJP, and SPEI were −2.66, −2.25, −87.15%, and −2.24, respectively, with corresponding evaluations of drought intensity levels as "extreme drought", "extreme drought", "severe drought", and "extreme drought") and FR (SPI, CPZI, PJP, and SPEI were −2.44, −3.05, −87.07%, and

−2.38, respectively, with corresponding evaluations of drought intensity levels as "extreme drought", "extreme drought", "severe drought", and "extreme drought"). Furthermore, during the same period, the drought intensity level assessed by the SPEI for XR ("severe drought") was higher than the drought severity levels assessed by the other three drought indices ("moderate drought"). On the other hand, the drought intensity level assessed by the PJP index for PYL ("moderate drought") was lower than the drought severity levels assessed by the other three drought indices ("severe drought"). On the whole, SPEI has different degrees of underestimation in the evaluation of drought intensity in typical months of each sub-basin, while PJP has the risk of overestimating regional drought intensity. SPI and CPZI have stable performance in the characterization of drought intensity levels of each sub-basin in 2003.

Figure 3 shows the representation of the selected four drought indices in describing the drought conditions during the typical year of 2019. According to the records of drought disasters in the PLB in 2019, there was a continuous drought throughout the summer, autumn, and winter seasons. In early August, the northern part of the basin showed initial signs of drought, which then rapidly developed and spread. From late September to mid-December, severe and higher-level drought conditions persisted in most areas for nearly three months. The total number of stations with severe drought or worse reached 7308, ranking first in history. Severe and higher-level drought conditions covering over 90% of the area of Jiangxi Province persisted for nearly three consecutive months, representing the largest drought extent.

Taking October 2019 as an example, in the DGR, XJR, and RR, the SPI, CPZI, and PJP show consistency in characterizing the intensity of regional drought, with ratings of "mild drought", "mild drought", and "moderate drought", respectively. However, the evaluation results of SPEI are "moderate drought", "moderate drought", and "severe drought", respectively, indicating a possibility of overestimating regional drought. During the same period, both SPI and CPZI indicate a drought intensity level of "mild drought" in the UGR, MGR, and FR. However, the evaluation results of PJP and SPEI are consistently "moderate drought". This suggests a possibility of overestimating regional drought with the PJP and SPEI. In addition, in November 2019, the drought intensity levels in the FR and XJR indicate consistency among the SPI, CPZI, and SPEI evaluations, all showing "severe drought". However, the PJP evaluation indicates a drought level one grade lower than the other three drought indices, rating it as "moderate drought". During the same period, the PJP evaluation indicates a drought intensity level of "severe drought" in the MGR, which is one grade higher than the mild drought intensity indicated by the other three drought indices. In the UGR, the SPI and SPEI evaluations show consistency in rating the drought level as "severe drought", while the CPZI and PJP evaluations indicate "moderate drought" and "extreme drought" levels, respectively, suggesting a risk of underestimating or overestimating regional drought. On the whole, SPEI, PJP, and CPZI have overestimated or underestimated the meteorological drought intensity of typical months in each sub-basin, and SPI has a stable performance in the characterization of the meteorological drought intensity of each sub-basin.

Based on the actual drought records in the typical years of 2003 and 2019, the selected four drought indices exhibit a high level of consistency in describing the trends and changes in drought conditions. Figure 4 presents a radar map of the probabilities of "mild drought" or higher occurring in each sub-basin at monthly, seasonal, and annual scales based on the selected four drought indices over the evaluation period from 1960 to 2020. From the figure, it can be seen that the evaluation results of SPI and CPZI are relatively close in each sub-basin, while the PJP overall underestimates the probability of drought occurrence in each sub-basin, and the SPEI overall overestimates the probability of drought occurrence in each sub-basin. At the monthly scale, SPI is similar to CPZI in describing the probability of drought occurrence in each sub-basin. At the seasonal and annual scales, SPI closely matches or even exceeds CPZI in describing the probability of drought occurrence in each sub-basin. Considering the perspective of drought warning and the designation of

drought response measures, it is recommended to use the SPI index to describe the drought conditions in PLB.

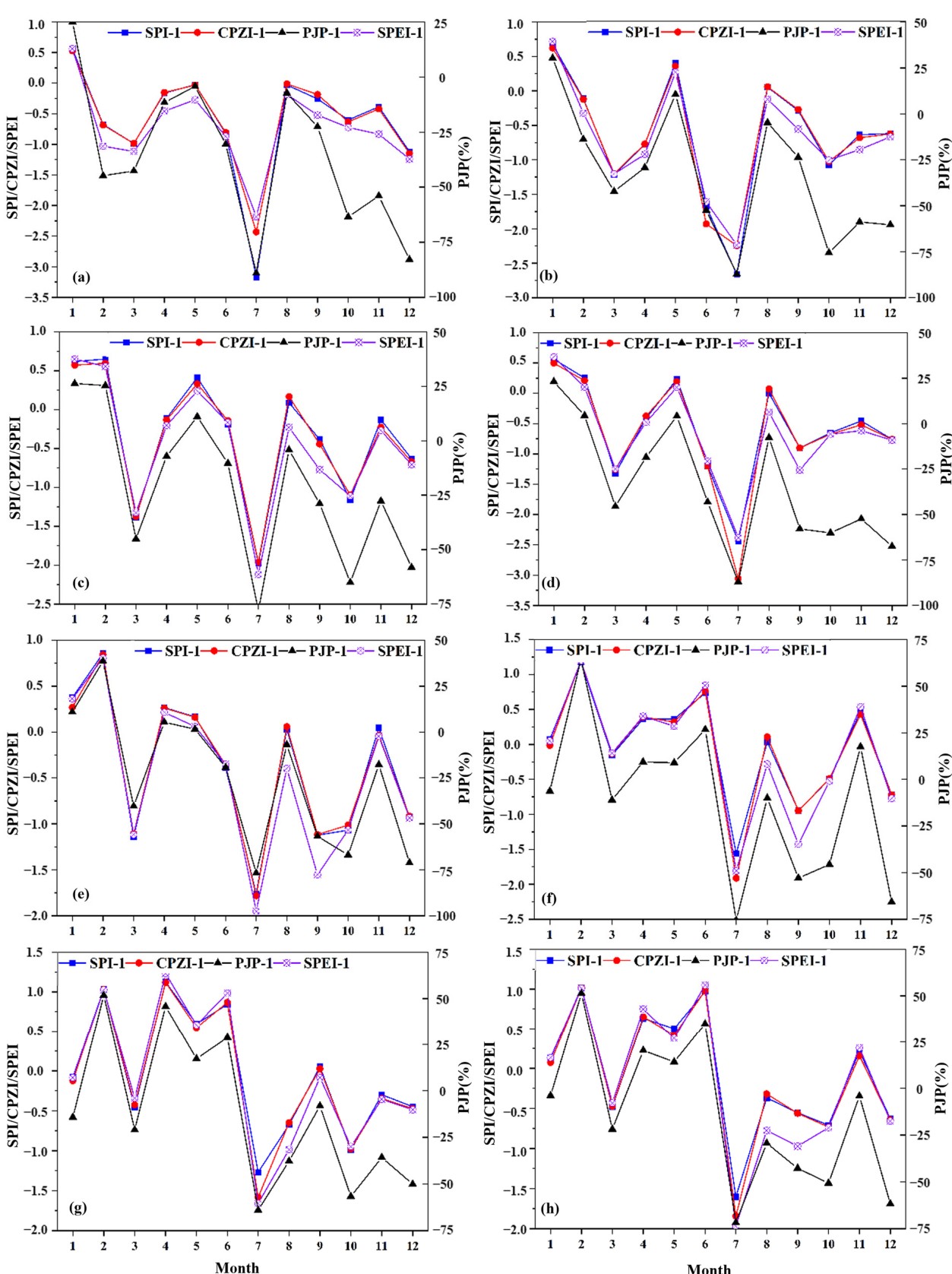

**Figure 2.** Comparison of drought index in each sub-basin in 2003: (**a**) UGR; (**b**) MGR; (**c**) LGR; (**d**) FR; (**e**) XJR; (**f**) RR; (**g**) XR; (**h**) PYL.

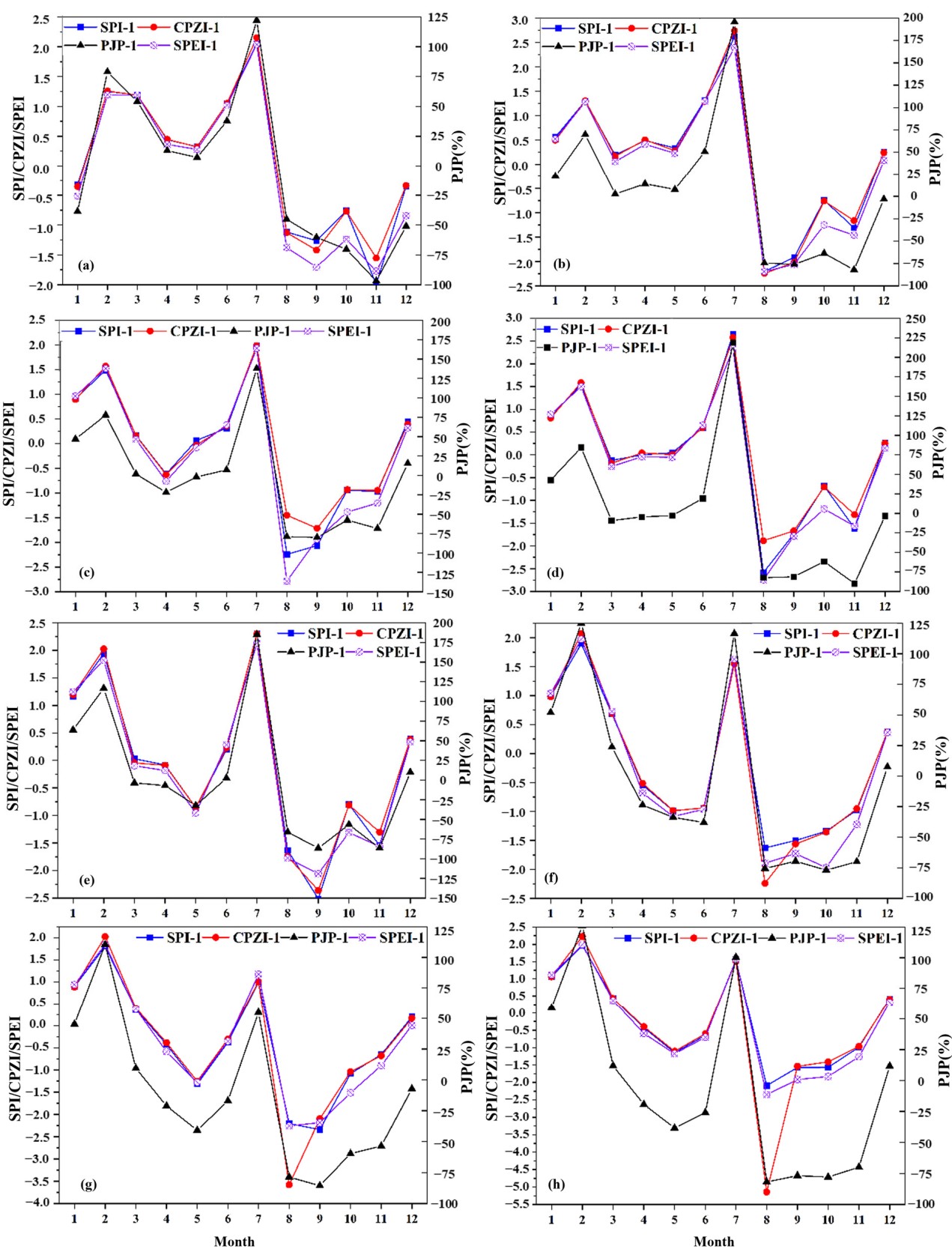

**Figure 3.** Comparison of drought index in each sub-basin in 2019: (**a**) UGR; (**b**) MGR; (**c**) LGR; (**d**) FR; (**e**) XJR; (**f**) RR; (**g**) XR; (**h**) PYL.

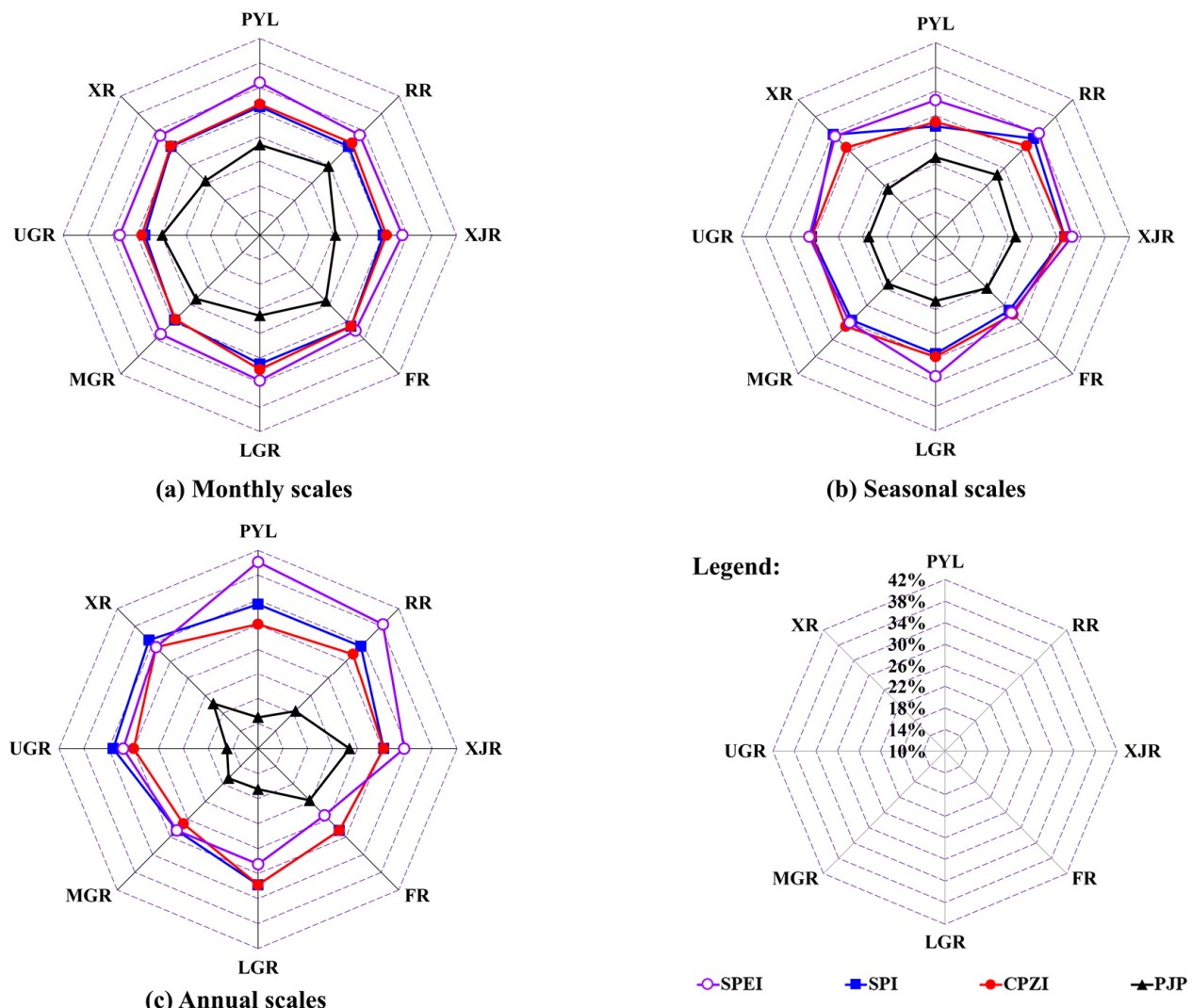

**Figure 4.** Radar map of drought occurrence probability at different time scales in each sub-watershed.

## 4. Temporal and Spatial Evolution Characteristics of Drought

*4.1. Frequency and Intensity*

Figure 5 presents the occurrence frequency of drought at different intensity levels in various sub-basins of PLB on an annual scale from 1960 to 2020. Here, the annual-scale drought frequency is defined as the percentage of the number of years (N = 60 years) with different drought intensity levels based on the SPI for February (SPI12) over the past 12 months, which is classified according to the criteria in Table 1. We can see that in the PLB, mild drought and moderate drought are the main occurrences, with an average probability of 25.2%. The probability of mild drought ranges from 13.11% to 22.95%, with the smallest being XJR and the largest being RR. The probability of moderate drought ranges from 3.28% to 11.48%, with the smallest being MGR and the largest being XJR and PYL.

Further, we analyzed in detail the spatial distribution of drought above severe severity in PLB at different time scales (year, season, and month), as shown in Figure 6. In terms of spatial distribution, severe drought and worse still occur most frequently in summer and autumn, followed by spring and winter. By setting the frequency of severe meteorological drought and worse at 10.00% as the area prone to drought, a detailed analysis was conducted on the areas prone to severe drought and worse in PLB, as well as the corresponding occurrence areas (as shown in Table 2).

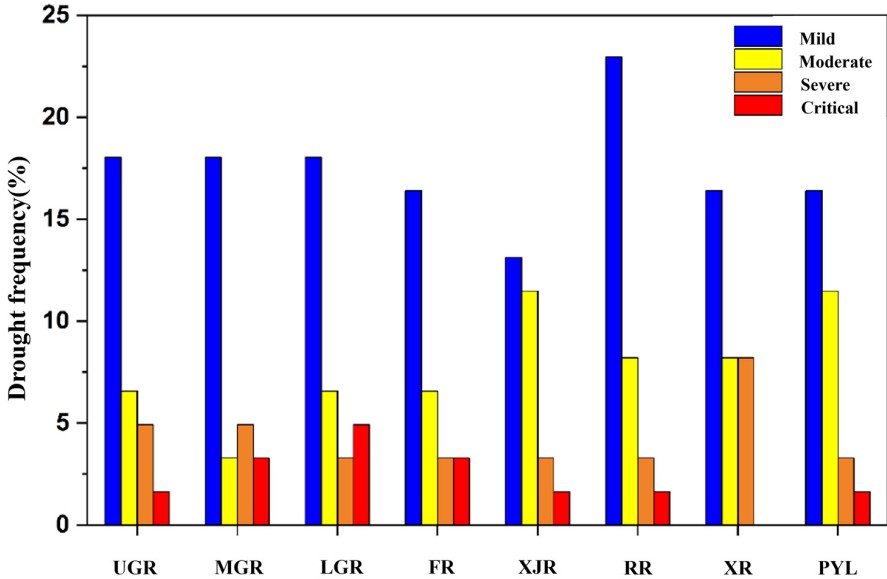

**Figure 5.** Annual frequency of drought in each sub-basin of Poyang Lake.

**Figure 6.** Spatial distribution of drought frequency above severe severity at different time scales in Poyang Lake Basin: (**a**) spring; (**b**) summer; (**c**) autumn; (**d**) winter; (**e**) annual scale; (**f**) monthly scale.

**Table 2.** Percentage of the area with severe drought frequency over different time scales in Poyang Lake Basin (%).

| Frequency (%) | Spring | Summer | Autumn | Winter | Annual | Monthly |
|---|---|---|---|---|---|---|
| 0–2.5 | 0.05 | 0.20 | 0.10 | 1.78 | 1.17 | - |
| 2.5–5.0 | 2.39 | 9.23 | 1.14 | 32.76 | 15.31 | 32.03 |
| 5.0–7.5 | 24.47 | 22.49 | 12.98 | 60.72 | 66.09 | 67.97 |
| 7.5–10.0 | 66.56 | 45.89 | 57.46 | 4.56 | 17.16 | - |
| 10.0–12.5 | 6.36 | 20.59 | 27.46 | 0.17 | 0.27 | - |
| 12.5–15.0 | 0.18 | 1.51 | 0.83 | - | - | - |
| 15.0–17.5 | - | 0.09 | 0.04 | - | - | - |
| Vulnerable area [1] | 6.54 | 22.19 | 28.33 | 0.17 | 0.27 | - |

Note: [1] Vulnerable area refers to the area where the frequency of severe drought is above 10.00%.

Severe drought and worse occur most frequently and have the widest regional distribution in summer and autumn. Among them, the area prone to severe drought and worse in summer accounts for 22.19% of PLB. It mainly occurs in the middle part of PLB, specifically in the entire Ganjiang River basin (GR), the upstream and middle reaches of FR, and some areas of XJR. The area prone to severe or worse droughts in autumn accounts for approximately 28.33% of PLB, which is larger in size and more widely distributed compared to the summer. The lower reaches of XJR, the upper reaches of XR, the UGR, the MGR, the FR, and some areas of PYL are susceptible to severe or worse droughts in autumn. In the study area, the area prone to severe drought in spring accounts for about 6.54% of the whole basin area, and it is scattered in the lower FR and GR. At the winter and annual scales, contiguous areas susceptible to severe drought are essentially absent within the basin.

### 4.2. Extent and Trend

To assess the occurrence of drought at different time scales in the PLB, the weighted average of drought indexes and their corresponding areas (calculated using the Thiessen polygon method) from 185 rainfall stations in the PLB were used to calculate the basin-wide average drought index. Figure 7 shows the statistical results of the range of drought occurrence over the past years at different time scales. At the monthly time scale, the PLB experienced extreme, severe, moderate, and mild widespread drought events 3, 24, 39, and 91 times, respectively, accounting for percentages of the total number of months as follows: 0.33%, 3.67%, 6.33%, and 14.67%. Among them, extreme drought occurred globally in February 1999 and September 2011, with corresponding SPI values of −2.583 and −2.068, respectively. March, November, and May were the months with the highest frequency of severe, moderate, and mild droughts occurring globally, with frequencies of 4, 8, and 10 times, respectively. At the seasonal scale, autumn had the highest frequency of global meteorological drought, occurring 15 times, followed by winter (13 times), while spring and summer both had 12 occurrences of global drought. At the seasonal scale, autumn had the highest frequency of regional drought, occurring 15 times, followed by winter (13 times), with both spring and summer experiencing 12 occurrences of regional drought.

In terms of the duration of regional meteorological drought, continuous droughts lasting more than four months occurred from June to September 1978, January to May 2011, February to May 2018, August to November 2019, and July to October 2022. There were consecutive droughts lasting for three months in the following periods: October to December 1979, April to June 1985, May to July 1991, September to November 1992, and September to November 1996.

This section employs the Mann–Kendall trend analysis method to analyze the spatiotemporal trends of drought in PLB. The drought variations at different time scales (annual and seasonal) in PLB are shown in Figure 8a–f. Taking summer as an example, most of the SPIs of each sub-basin (177 sites) in summer showed an upward trend (46 sites with a significant upward trend), accounting for 95.68% of the total number of sites in

the whole basin. This indicates that the summer drought conditions in each sub-basin have shown an overall mitigating trend over the past 50 years. In the entire basin, there are 150 stations showing a decreasing trend in SPI in spring (with 1 station exhibiting a significant decreasing trend) and 38 stations showing a non-significant increasing trend. This indicates an overall worsening trend of meteorological drought conditions in various sub-basins over the past 50 years in spring. There are 119 stations showing an increasing trend in SPI in autumn (with 1 station exhibiting a significant increasing trend) and an additional 69 stations showing a non-significant decreasing trend. This indicates an overall reduction in meteorological drought in various sub-basins during the autumn season. The UGR and the downstream of the XJR are typical regions where drought is intensifying in autumn. There are 149 stations showing an increasing trend in SPI in winter (with 5 stations exhibiting a significant increasing trend) and an additional 39 stations showing a non-significant decreasing trend. The sub-basins of PLB show an overall reduction in drought during the winter season, while certain regions in the UGR and LGR exhibit an increasing trend in winter drought. At the annual scale, there are 156 stations in the entire basin showing an increasing trend in SPI (with 7 stations exhibiting a significant increasing trend) and an additional 33 stations showing a non-significant decreasing trend. The sub-basins of PLB show an overall reduction in drought at the annual scale, while the UGR exhibits an increasing trend in drought at the annual scale. Taking October as an example, among 185 sites, 181 sites showed a worsening trend of drought, and 30 sites showed a significant worsening trend, accounting for 97.84% and 16.22% of the sites, respectively. This means that although the overall drought in autumn is showing a slowing trend, it is still a season prone to drought, and the drought severity in October may worsen.

*4.3. Development Path*

According to the above studies on drought frequency and extent, this section selects 2022 as a typical year to analyze the drought characteristics of the aforementioned year. Building upon this analysis, we will examine the development path of drought in a typical year.

Figure 9 shows the spatial distribution of drought intensity for each month in the year 2022. As can be seen from the figure, July–October is a continuous dry period, which belongs to the sudden drought of the whole basin. From January to June 2022, the average precipitation in the PLB was 1229.5 mm, which was 1.12 times the annual average of 1097.5 mm in the same period. From July to October, the average precipitation in the whole basin was 120.9 mm, which was only 27.0% of the annual average of 448.5 mm in the same period. The total precipitation in July (80.5 mm) accounted for 51.8% of the annual average (156.1 mm). The total precipitation in August (27.9 mm) was 19.7% of the annual average of 141.5 mm. The total precipitation in September (3.1 mm) was only 3.6% of the annual average (87.3 mm). The total precipitation in October (9.38 mm) was only 14.8% of the annual average (63.6 mm). Table 3 shows the percentages of the area affected by drought of varying intensity levels in the Poyang Lake Basin in the typical year of 2022. The drought area during this period ranged from 62.9% to 99.99% of the basin area, with an average of 87.56%. Combining Figure 9 and Table 3, it can be concluded that a sudden onset of drought occurred across the entire basin, with mild and moderate drought being the primary intensities in July 2022. The drought center is concentrated in the MGR. In August, the drought rapidly escalated, with the affected area expanding to almost the entire basin. The drought intensity was mainly moderate and severe, and multiple drought centers appeared throughout the basin. In September, the drought continued to intensify, with extreme drought being predominant across the entire basin. In October, the drought center returned to the MGR, with severe drought being the main intensity across the basin. In November, the average precipitation in the entire basin was about 177.47 mm, leading to a rapid alleviation of the drought situation.



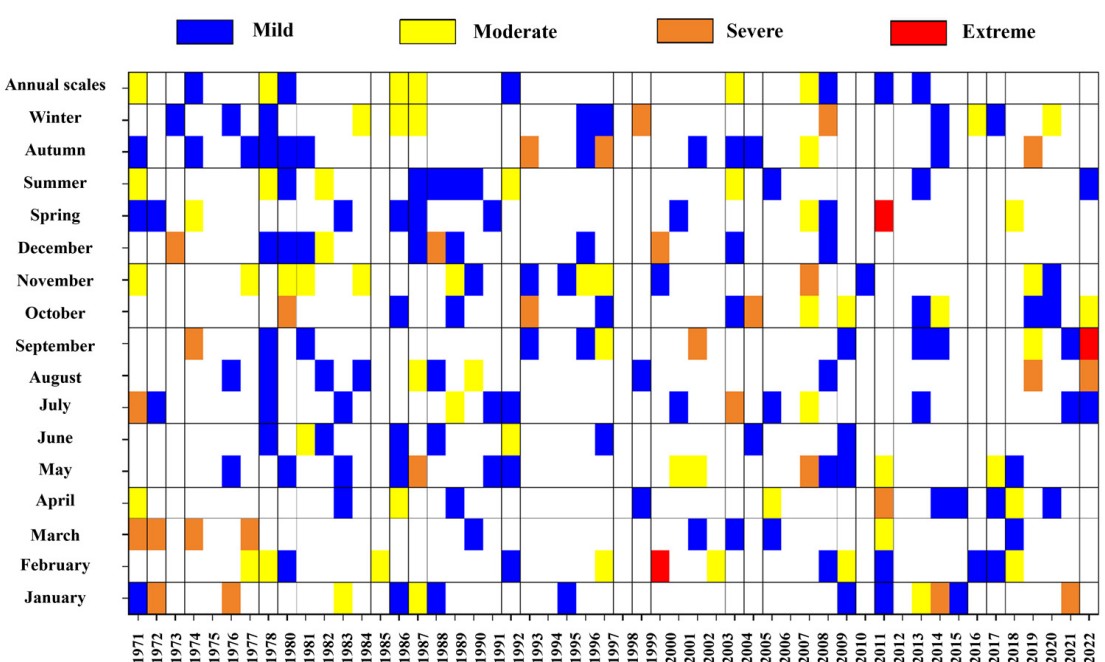

**Figure 7.** Assessment results of regional meteorological drought in Poyang Lake at different time scales.

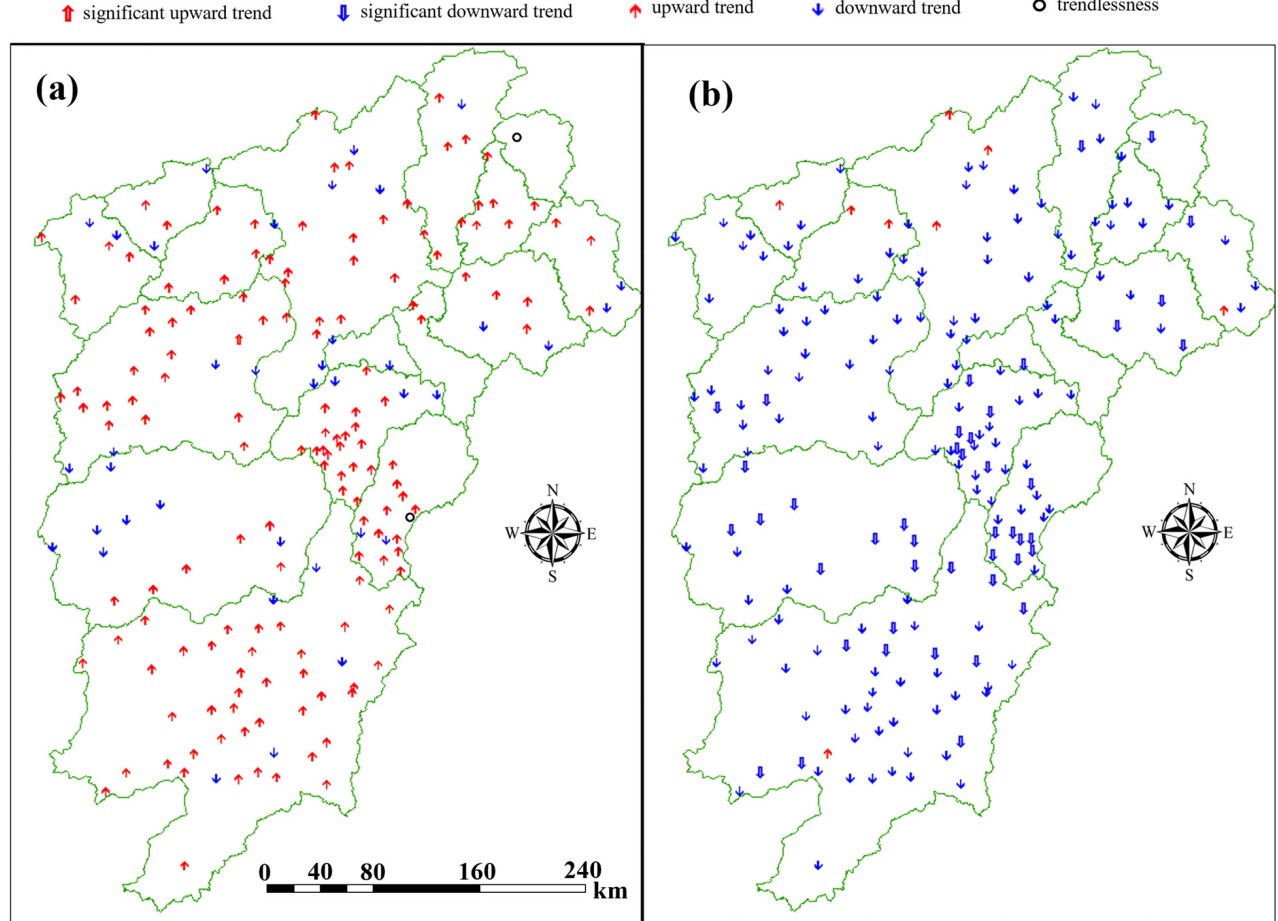

**Figure 8.** *Cont.*

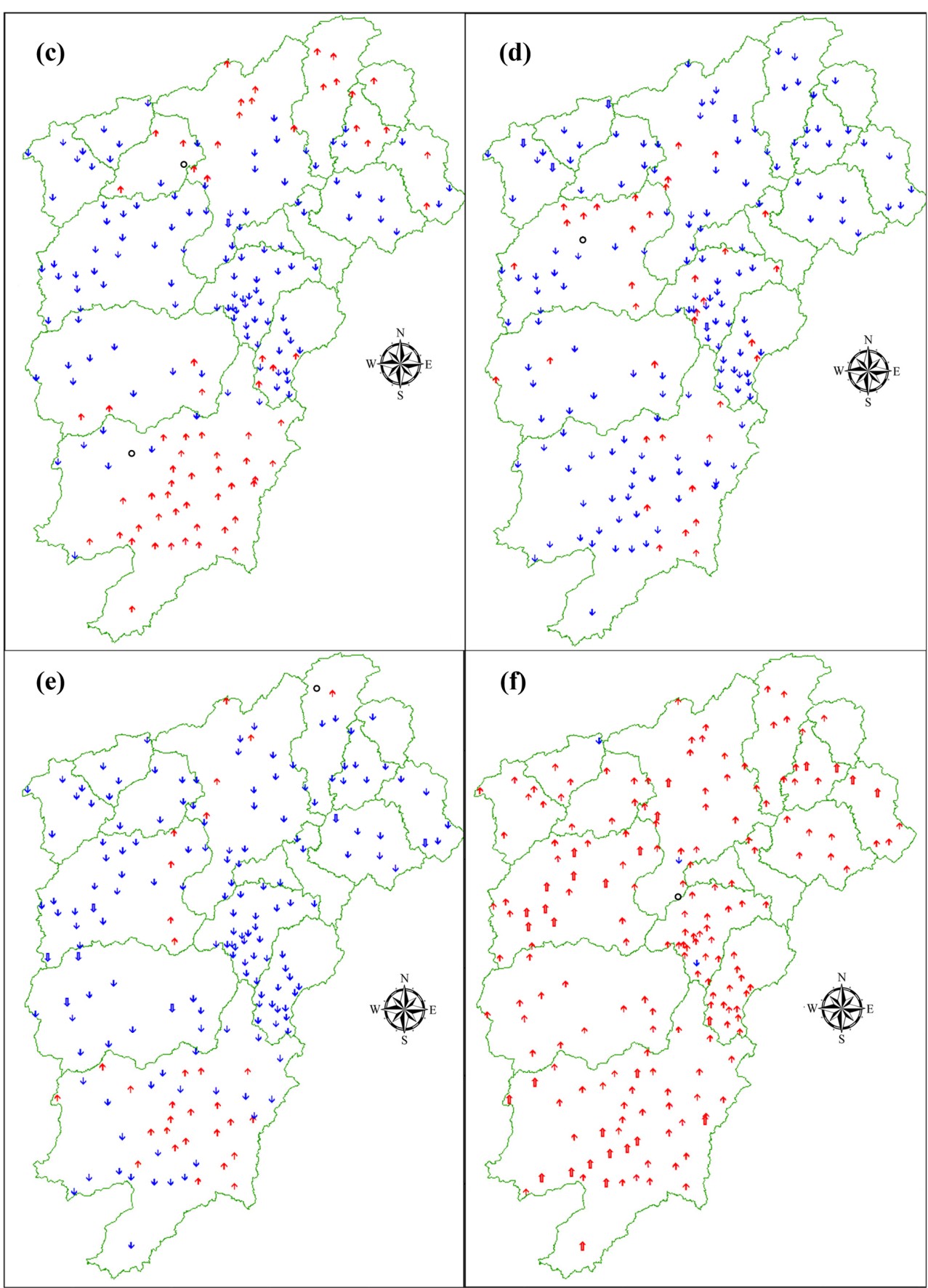

**Figure 8.** Drought trends at different time scales in Poyang Lake Basin: (**a**) spring; (**b**) summer; (**c**) autumn; (**d**) winter; (**e**) annual scale; (**f**) monthly scale.

**Table 3.** Percentage of drought area with different intensity levels in Poyang Lake Basin Area in typical year (2022) (%).

| Month | Mild Drought | Moderate Drought | Severe Drought | Extreme Drought | Moderate Drought or Worse | Severe Drought or Worse |
|---|---|---|---|---|---|---|
| 1 | 0.09 | 0 | 0 | 0 | 0 | 0 |
| 2 | 0.10 | 0.07 | 0.07 | 0.01 | 0.15 | 0.08 |
| 3 | 2.76 | 0 | 0 | 0 | 0 | 0 |
| 4 | 6.47 | 4.12 | 5.51 | 0.62 | 10.25 | 6.13 |
| 5 | 8.72 | 2.57 | 0.31 | 0 | 2.89 | 0.32 |
| 6 | 1.76 | 0.06 | 0 | 0 | 0.06 | 0 |
| 7 | 32.78 | 21.22 | 5.09 | 3.80 | 30.11 | 8.89 |
| 8 | 1.97 | 16.99 | 41.41 | 39.43 | 97.83 | 80.84 |
| 9 | 0.06 | 1.67 | 16.91 | 81.36 | 99.93 | 98.26 |
| 10 | 19.82 | 41.65 | 36.47 | 0.84 | 78.12 | 37.31 |
| 11 | 0 | 0 | 0 | 0 | 0 | 0 |
| 12 | 2.90 | 0 | 0 | 0 | 0 | 0 |

Figure 10 provides a detailed illustration of the spatial propagation direction and development path of drought from July to October 2022. In the figure, the arrows represent the development path of the drought, the numbers indicate the drought intensity values in each month, the direction of the arrows represents the migration direction of the drought center, and the solid red and dashed blue lines, respectively, indicate the increase and decrease in drought intensity during the migration process. The spatial distribution of the average SPI values for each month in the 2022 drought event is presented using the spatial inverse distance weighting (IDW) interpolation method. Based on Figure 9, it can be observed that the center of this drought is located in the UGR. The intensity of the drought center is approximately −4.53. By August 2022, the drought had rapidly developed into multiple drought centers throughout the entire basin, including the LGR, the upper reaches of the FR, the middle reaches of the XJR, and the lower reaches of the XR. The intensity of the drought centers ranged from −3.885 to −4.71. The drought centers in September and October overlapped. The drought center returned to the UGR and MGR in October, with intensities of −5.092 and −2.748, respectively.

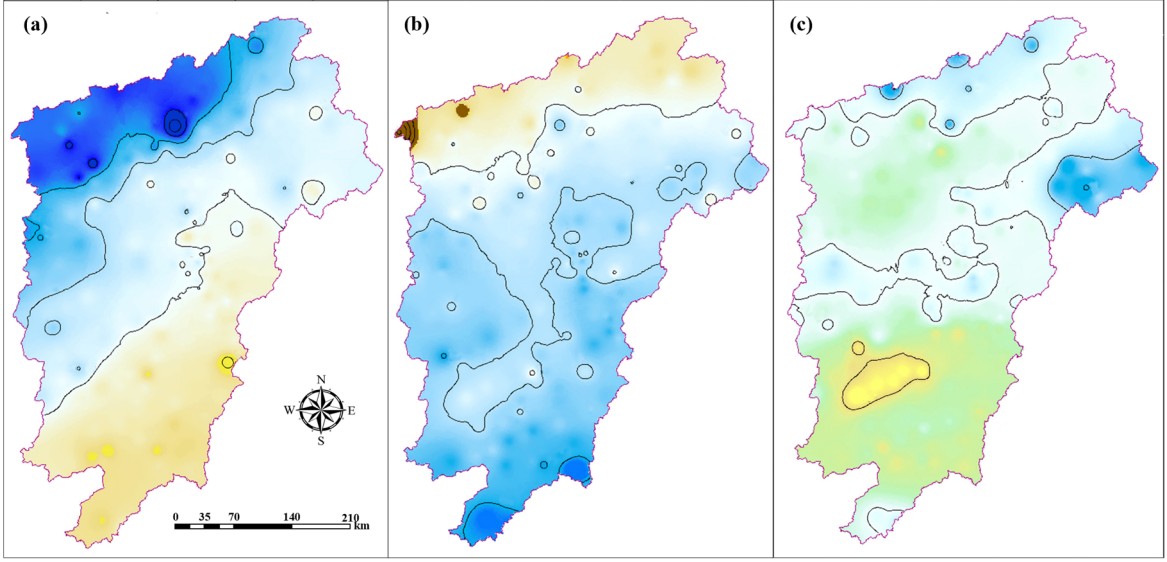

**Figure 9.** *Cont.*

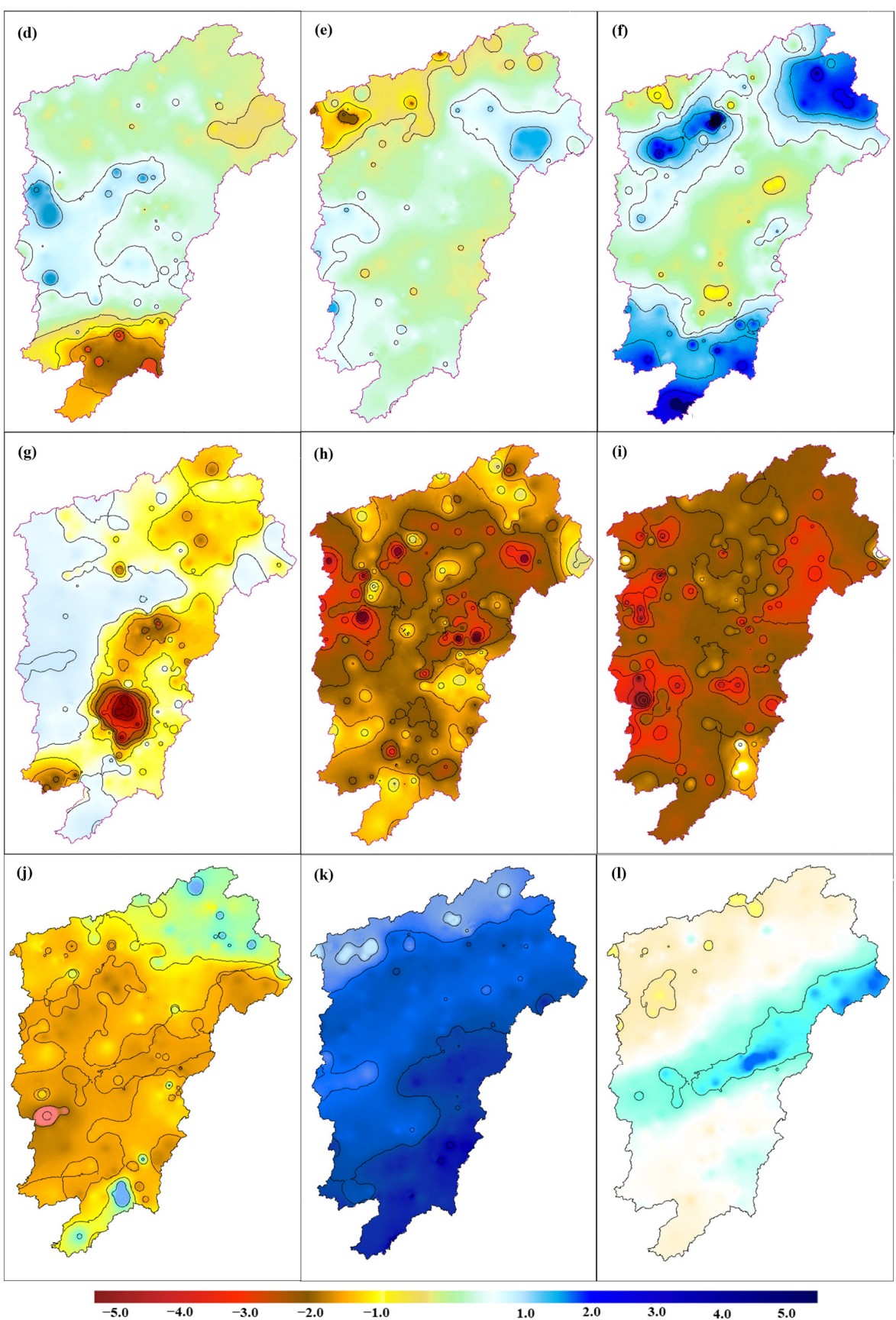

**Figure 9.** Temporal and spatial distribution characteristics of meteorological drought in Poyang Lake Basin in 2022: (**a**) January; (**b**) February; (**c**) March; (**d**) April; (**e**) May; (**f**) June; (**g**) July; (**h**) August; (**i**) September; (**j**) October; (**k**) November; (**l**) December.

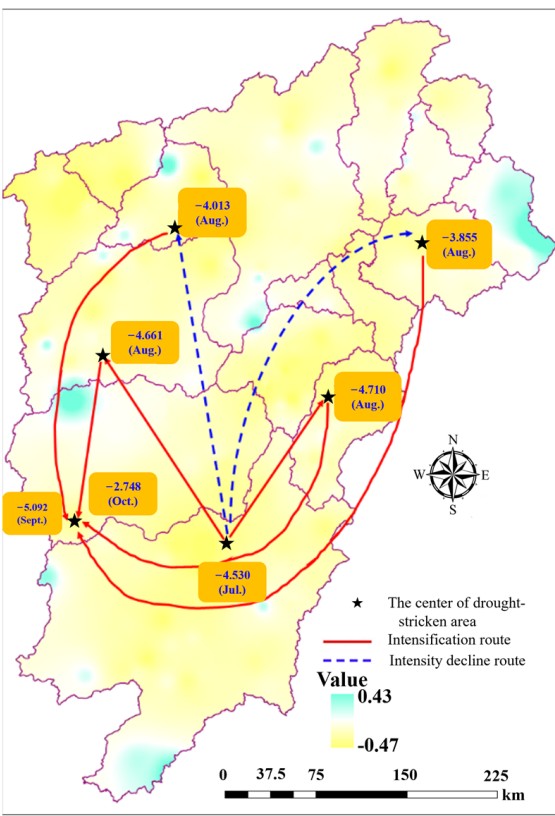

**Figure 10.** Development path of drought in a typical year (2022).

## 5. Discussion

PLB is one of the most climate-sensitive areas in the Yangtze River Basin, especially since the 1990s when there has been a significant increase in the frequency of extreme precipitation events [41]. In this study, the estimation of SPI for drought reflects the impact of precipitation very well, and the research results are not significantly different from those of previous scholars. In terms of indicator applicability, among the four widely used drought indexes in China, SPI can better characterize the drought characteristics of the PLB. This finding is consistent with previous studies, where it has been observed that SPI performs better at shorter time scales when the regional area is less than 1,000,000 km$^2$ [40,42]. Although SPI only considers precipitation, it can effectively identify drought processes in the PLB under current climate change, similar to other complex multi-factor drought indices [43]. In terms of frequency and intensity, droughts are more frequent in the autumn and summer in sub-basins of PLB, with the FR, LGR, and middle reaches of the XJR being prone to severe drought. This finding is consistent with other research results. Ye [41] found that the occurrence rate of autumn drought in the Poyang Lake Basin is the highest; Zheng [35] found an increase in drought frequency in the summer and autumn in the PLB; Hong [33] identified the LGR, middle and lower reaches of the FR, and the middle reaches of the XJR as the most vulnerable areas to severe drought intensity. In terms of trends, the overall trend of spring drought in the PLB is increasing, while there is an overall decreasing trend in summer, autumn, winter, and annual scales. This is similar to other research findings. Liu [28] found a drought trend in the spring and an overall wet trend in other seasons in the PLB. Hong and Wang [33,34] found that drought in the Poyang Lake Basin exhibits clear seasonal characteristics, with more pronounced drought trends in the spring and autumn seasons. The inconsistency between this study and other findings may be due to the consideration of other influencing factors (such as temperature, soil, etc.). Therefore, understanding the spatiotemporal changes of drought and its influencing factors is of great significance for mitigating and preventing natural disasters.

Studies have shown that the Western Pacific Subtropical High (WPSH) is one of the most important circulation systems affecting the weather and climate in East Asia [44]. China's geographical location determines that it is greatly affected by it. The change in its location and area (intensity) has an important effect on the precipitation during flood season in China, and its interannual change determines the occurrence of drought and flood in eastern China [45]. In recent years, severe weather has occurred frequently in the Poyang Lake Basin, such as the catastrophic flood in the summer of 1998 and the continuous hot summer weather in the middle and late July of 2003 [46]. These abnormal weather and climate events are closely related to the subtropical high activity.

We conducted a systematic analysis of five typical years (1978, 2003, 2007, 2011, 2013, 2019, and 2022) to investigate the correlation between the center location (longitude and latitude) and SPI of drought, as well as the Western Pacific Subtropical High area index (AI), intensity index (II), ridge position index (RPI), and westward ridge point index (WRPI). As shown in Figure 11, the SPI of the drought center is significantly negatively correlated with the AI, II, and RPI. This is consistent with the circulation condition that the West Pacific Subtropical High is northward and the Pacific–Japan Pattern anomaly is drought.

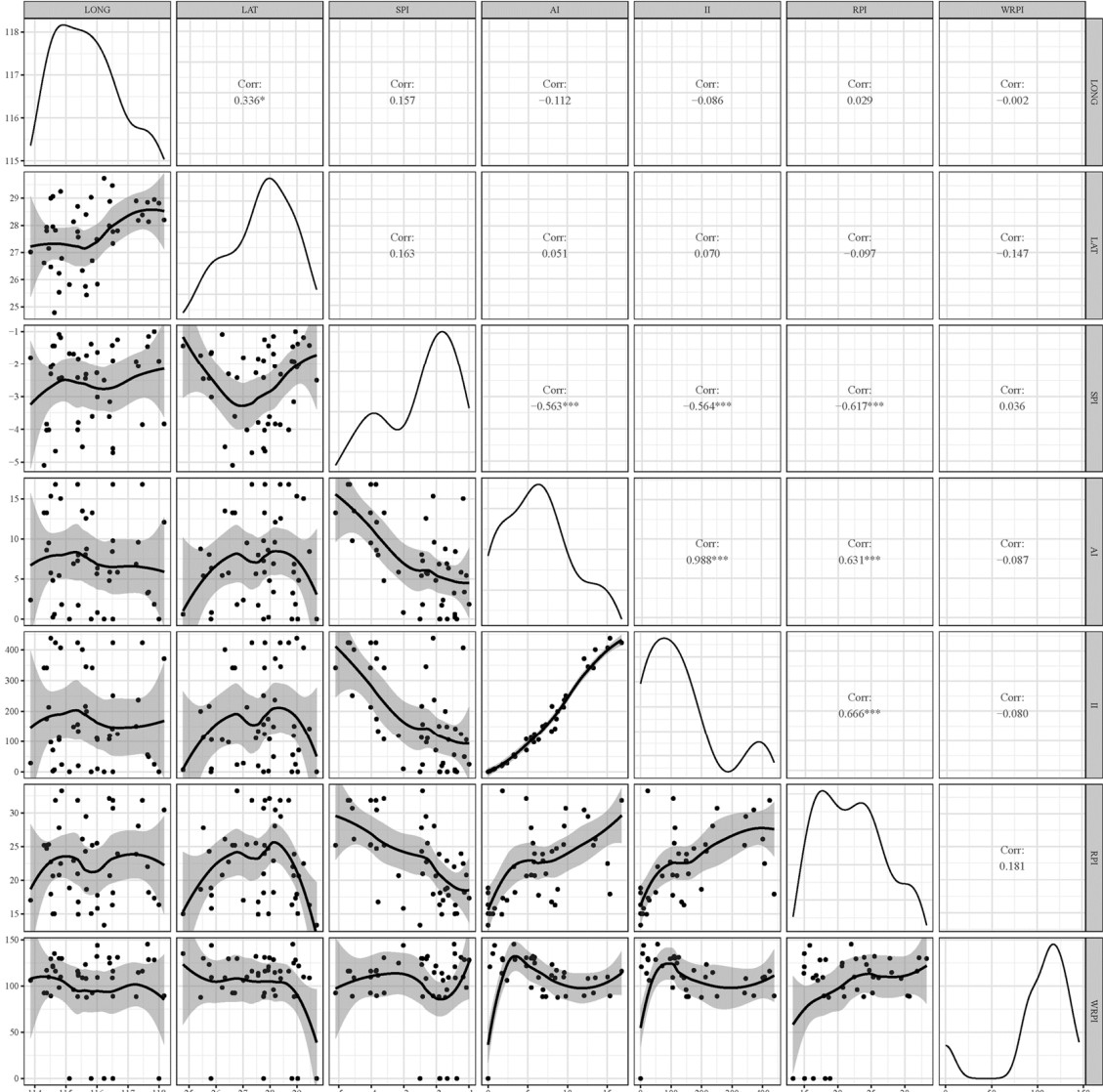

**Figure 11.** Correlation of drought center location, SPI, and WPSH index in typical years (The asterisk of the correlation coefficient indicates that the correlation is more significant. "*" represents $p < 0.05$, and "***" represent $p < 0.001$.).

From the above analysis, it can be seen that there are differences in the spatiotemporal distribution patterns of drought among sub-basins of PLB. Based on the different characteristics of drought, as described earlier, we categorize the sub-basins into three types. The first category is high-frequency drought areas, including the GR, PYL, upper and middle reaches of the XR, upper and middle reaches of the XJR, and middle and lower reaches of the RR. The second category is severe drought-prone areas, including the FR, LGR, and the middle reaches of the XJR. The third category is areas with worsening drought conditions, including the UGR and the lower reaches of the XJR. In view of the above three types of risk areas, this paper proposes corresponding drought response policies and key measures. For high-frequency drought-prone areas, the main focus is on enhancing and improving the drought monitoring network, establishing a system for regional drought forecast, early warning, rehearsal, and pre-planning. For severe drought-prone areas, the construction of emergency water sources should be promoted, and the early warning index of water level should be established on the basis of the areas with high frequency of drought. For areas with worsening drought conditions, building upon the foundations of the previous two categories, efforts will be made to advance the construction of large and medium-sized reservoirs, connect small and micro-irrigation areas into continuous blocks, and enhance irrigation district and water-saving transformations.

## 6. Conclusions

In this study, PJP, SPI, CPZI, and SPEI were selected to identify and optimize the most suitable drought indexes for drought research in PLB through the comparison of drought indices in typical years. On this basis, a systematic study was conducted on the spatiotemporal evolution patterns of drought in the PLB, including drought intensity and frequency, trends, and the extent of drought. Furthermore, a detailed analysis of the characteristics and development pathway of the drought in the lower region in 2022 was conducted. The main conclusions are as follows:

(1) The PJP, SPEI, and CPZI tend to overestimate or underestimate the drought intensity levels in the sub-basins of the study area in typical years. On the other hand, the SPI demonstrates a relatively robust performance. Considering the perspective of drought warning and the formulation of drought mitigation strategies, we recommend using the SPI to describe the meteorological drought conditions in the Poyang Lake Basin;

(2) In terms of time scale, droughts occur most frequently and over the widest range of regions in summer and autumn, followed by spring and winter. Among them, the area prone to severe droughts in summer accounts for 22.19% of PLB's total area. The area prone to severe droughts in autumn is approximately 28.33% of PLB's total area. In terms of spatial scale, RR is the sub-basin with the highest frequency of drought occurrence in PLB, mainly experiencing mild droughts, while severe droughts and worse occur more frequently in GR and XR;

(3) In PLB, the overall trend of spring droughts is worsening, while summer, autumn, winter, and annual-scale droughts show an overall alleviating trend. The UGR and the lower reaches of the XJR are typical regions where autumn droughts are worsening. In some areas of the UGR and LGR, winter droughts are worsening, and there is a worsening trend in annual-scale droughts in the UGR;

(4) This study presented the characteristics and evolution of drought development paths in the PLB during a typical drought year in 2022. It is found that a negative correlation between the SPI of the drought center and the AI, II, and RPI of the WPSH.

In summary, the results of this study can provide a scientific basis and support for future drought control work and promote the implementation of rational and effective management of water resources in the basin. In addition, since there are many factors affecting the variation of drought characteristics in PLB, the changing trend and its mechanism under the influence of single or multiple factors will be the focus of future research.

**Author Contributions:** Conceptualization, Z.G. and F.G.; methodology, H.B. and Z.G.; software, Z.G.; validation, H.Q. and S.W.; formal analysis, Z.G and H.Q.; investigation, H.B. and F.G.; resources, F.G.; writing—original draft preparation, Z.G.; writing—review and editing, F.G.; visualization, B.Z. and Z.G. All authors have read and agreed to the published version of the manuscript.

**Funding:** This research was jointly supported by the Natural Science Foundation of Zhejiang Province (No. LZJWY23E090009) and the Natural Science Foundation of Jiangxi Province Youth Fund project (No. 20232BAB214089).

**Data Availability Statement:** The data that support the findings of this study are openly available at www.jxssw.gov.cn (accessed on 15 March 2023).

**Conflicts of Interest:** The authors declare no conflicts of interest.

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
