# Peer review of "The Applicability of the Drought Index and Analysis of Spatiotemporal Evolution Mechanisms of Drought in the Poyang Lake Basin"

_water, doi:10.3390/w16050766_

Round 1

Reviewer 1 Report

Comments and Suggestions for Authors

This study deals with the applicability of drought index in the spatiotemporal monitoring of drought conditions and its characteristics in a lake ecosystem. The spatiotemporal evolution gives some useful insights into integrated water resources management. The main points for revision and improvement are listed below:

·         The abstract provides a succinct overview of the methodology employed by the authors. However, it would greatly benefit from the inclusion of specific numeric results and outputs derived from their analysis. This addition would enhance the comprehensiveness of the abstract and provide readers with a clearer understanding of the study's findings.

·         In line 34, it is recommended to incorporate official definitions of drought and distinguish its nature from other natural disasters. Including statements from reputable sources such as the World Meteorological Organization (WMO) and the Centre for Research on the Epidemiology of Disasters (CRED) would enhance the clarity of the manuscript. For instance, the authors should specify that drought is a meteorological phenomenon characterized by a prolonged dry period in the natural climate cycle, resulting in water shortages. It is crucial to emphasize that drought, unlike other natural hazards such as floods, wildfires, and earthquakes, is a slow-onset hazard that becomes evident as societies and the environment begin to experience its impacts. Furthermore, highlighting the non-structural and extensive geographical impacts of drought would strengthen the manuscript (https://cred.be/sites/default/files/2021_EMDAT_report.pdf)

·         Line 77. At this juncture in the text, it is crucial to emphasize recent findings indicating that temporal variations of SPEI indices show differential reactions across time scales. Notably, shorter time scales (SPEI3 and SPI6) exhibit an increased frequency of drought events accompanied by a decrease in their duration. This suggests a recurrence of climatological phenomena for drought at these shorter time scales. In contrast, longer time scales (SPEI12 and SPEI24) experience less frequent drought episodes but with longer duration. Highlighting these temporal variations is essential for a comprehensive understanding of the dynamic nature of drought events (https://doi.org/10.3390/hydrology10080167)

·         Line 128. Add a basemap as background in Figure 1.

·         Line 227. A map with the location of the stations used should be added.

·         Line 323. You should add a table briefly illustrating the results such as mean duration, number of episodes and relative frequency of drought events.

·         Line 341. Labels font in figure 7 should be increase. In the current form are not readable (the same for figure 8 and figure 9).

·         Line 451. Please explain in the legend of the figure 10 what the redline represents.

·         The discussion section is notably absent from the article. It is crucial to include a comprehensive discussion that not only presents your findings but also engages in a comparative analysis, linking your outputs with relevant published research works. This addition will strengthen the overall interpretation of the results and contribute to the scientific discourse in the field.

Reviewer 2 Report

Comments and Suggestions for Authors

Although the article is interesting because of its topic, the following edits are required for publication in Water.

Line 70: What is SPEI index? What does this acronym stand for?

Line 86-96: Here the difference of the study from previous studies is discussed. However, this is not very satisfactory. You need to mention the shortcomings of previous studies in the previous paragraph. The last two paragraphs of the Introduction should be rewritten.

Line 103: km2 (Superscript)

Line 127: The legend is missing.

Line 139: You used SPEI in Line 70. Write abbreviations carefully. 

I suggest a review of all abbreviations in the article.

Line 207: Who set the rought index standards? Did you set them or are they commonly used values in the literature? If you set them, you should explain the reasons in detail. If they are in the literature, you should give a citation.

Line 228, 260: Vertical scales in Figure 2 and Figure 3 are expected to be the same. This makes it easier to understand the changes of different indices and basins.

Line 341: The resolution is bad. Please increase the DPI.

Line 400: Resolution is bad. Please increase DPI.

Line 419: Resolution is bad. Please increase DPI.

Although the results of the study are well explained, the ability to compare with other studies is poor. Therefore, a 'discussion' section should be added to the study and all aspects should be compared with other studies in the literature. 

Reviewer 3 Report

Comments and Suggestions for Authors

There are several points that the author should take into account.

1. The text of the article contains a lot of infographics showing changes in various parameters. Are the images presented taken from any sources or obtained by the author independently? If the images are copyrighted, what software was used?

2. The work makes an attempt to select the optimal set of methods to solve the problem. What are the optimality criteria? If this is the main result of the work, then it needs to be more clearly identified in the text.

3. It is not entirely clear what exactly is the result of the work. If it's a review article, then maybe there's a scientific point to it, but if not, then there must be some kind of result obtained.

4. The presented topics are relevant for many countries and territories, however, the author’s references are based mainly on the experience of Chinese colleagues. Perhaps studying and analyzing the experience of scientific teams in other countries would be useful.

Comments on the Quality of English Language

 Minor editing of English language required

Reviewer 4 Report

Comments and Suggestions for Authors

The paper deals with application of drought indices for describing the drought characteristics. The drought of the Poyang Lake Basin is of great practical significance for the people for this region.

The authors declared that the aim of this study was to provide a scientific basis for understanding the regional characteristics and evolution of meteorological drought in Poyang Lake Basin under changing environmental conditions, conducting a reasonable assessment of the impacts of meteorological drought, and formulating prevention strategies.

Thought this research objectively selects appropriate drought indices for characterizing drought features based on historical facts and describes and classifies different types od drought the last part of the aim has been not achieved. There are no any prevention strategies described in this paper. That is why I suggest to change the aim of this studies or to add needed information to the text body.

The paper has been well written, and properly planned. The obtained results were clearly described and the conclusions have been bases on the results.

There are minor mistakes like no reference to Figure 1 and Table 3 in the text, and no Table 4 to which the authors refer in the text (although the description shows that it is rather Table 3).

I would suggest also to increase the resolution of the Fig. 4 because the numbers and the words are almost illegible . The same problem is with the Fig 8. – the arrows indicating the upward or downward  trend and well as the significant ones are very hard to distinguish in that pictures.

I leave small editorial errors such as lack of space between numbers and units to the editor.

Round 2

Reviewer 1 Report

Comments and Suggestions for Authors

The article is accepted in the current form.

Author Response

Dear Reviewer,

Thank you for taking the time to review my paper and for providing valuable comments. Your suggestions have helped us to improve the quality of our work, and I am grateful for your assistance.

Thank you again for your support and guidance.

Sincerely,

Zihan Gui

Reviewer 2 Report

Comments and Suggestions for Authors

Most of the suggested corrections to the manuscript have been made. However, I do not see any improvements on the figures in the manuscript. Please make sure that all figures are easily readable (increase DPI).

Author Response

Dear Reviewer,

Thank you for your kind comments.We have separately uploaded the repackaged images after re-editing. Inserting the modified images into the article has made the file too large, causing some difficulties with the upload system.

Sincerely,

Zihan Gui